# Management of Individual Patient Expectations When Starting with Highly Effective CFTR Modulators

**DOI:** 10.3390/jpm11080811

**Published:** 2021-08-19

**Authors:** Bente L. Aalbers, Inez Bronsveld, Regina W. Hofland, Harry G. M. Heijerman

**Affiliations:** Department of Pulmonology, University Medical Center Utrecht, Heidelberglaan 100, 3584 CX Utrecht, The Netherlands; i.bronsveld@umcutrecht.nl (I.B.); r.w.hofland@umcutrecht.nl (R.W.H.); h.g.m.heijerman@umcutrecht.nl (H.G.M.H.)

**Keywords:** cystic fibrosis, CFTR modulators, emerging therapies, patient expectations, corrector, potentiator

## Abstract

Highly effective CFTR modulators such as elexacaftor/tezacaftor/ivacaftor (ELE/TEZ/IVA will become available for an increasing number of people with cystic fibrosis (pwCF) in the near future. Before the start of this therapy, many questions may arise concerning the expected effects. We assembled the currently available data from the literature about ELE/TEZ/IVA that focused on commonly asked questions from patients. Overall, the literature so far presents a very hopeful prospect of effects, not only on lung function, but also on nutritional status, sinonasal symptoms and quality of life. The effects in patients with pwCF with severe lung damage are also favorable. Treatment is generally well tolerated. In some cases, patient-derived cell models can help in predicting the effects for individual patients.

## 1. Introduction

In cystic fibrosis (CF), a mutation in both alleles of the CFTR gene leads to the production of a CFTR protein that is insufficient in structure, length, quantity or stability, depending on the exact mutation. As the protein normally functions as a chloride and bicarbonate channel across the cell membrane on the apical side, an insufficiently functioning CFTR protein in turn leads to a lack of chloride and bicarbonate transport, causing problems of thick mucus and pH imbalance in multiple epithelial tissues, hereby affecting many organs including the lungs, intestines, pancreas and sweat glands [1].

Although CF is a genetic disease, the most impactful developments in recent years were not therapies targeting the defective gene, but rather the resulting protein [2,3,4,5]. For an increasing number of mutations, this leads to a significant improvement in protein function. The class of small molecule drugs targeting the CFTR protein in order to restore functionality is called the CFTR modulators. The first to become available was the potentiator molecule ivacaftor, which acts on the CFTR protein affected by gating mutations in which it increases open channel probability, thus restoring the protein’s function [6,7,8]. It is used for CFTR mutations such as G551D and S1251N [9].

A second CFTR treatment option is a combination of the mentioned ivacaftor with a corrector, lumacaftor. Lumacaftor acts on protein processing in the celll leading to a larger proportion of the produced protein reaching the cell membrane instead of being degraded, where ivacaftor can then enhance the channeling function. This combination treatment is suitable for a larger number of people with CF (pwCF), as all pwCF homozygous for F508del are eligible; however, it is much less effective compared to ivacaftor for pwCF with suitable mutations [10]. A newer combination, tezacaftor/ivacaftor, was later added as a treatment option with similar efficacy to lumacaftor/ivacaftor, with slightly less side effects and less interactions with other drugs [11].

New hope arises with the emergence of triple therapy, of which the first available combination is elexacaftor/tezacaftor/ivacaftor (ELE/TEZ/IVA, Vertex therapeutics^®^). In phase III trials, this treatment was shown to have a drastic positive effect on clinical parameters such as lung function, pulmonary exacerbations, CFQ-R and sweat chloride concentration [12,13]. ELE/TEZ/IVA is currently accepted in the market for the treatment of patients with at least one F508del allele. In vitro studies, however, have also shown that this combination drug improved CFTR function in selected non-F508del genotypes [14,15,16,17].

However, with this new hope, many questions will arise concerning individual expectations surrounding the start of therapy. This review will not be able to address all of the related insecurities encountered in the CF clinic, but it aims to provide an overview of the current evidence concerning some frequent and relevant questions about expectations of the effects of newly started CFTR modulators. For many patients, this new modulator will be ELE/TEZ/IVA, sometimes switching from treatment with LUM/IVA (lumacaftor/ivacaftor), TEZ/IVA (tezacaftor/ivacaftor) or IVA (ivacaftor).

## 2. Questions to Address

### 2.1. My Lungs Are Severely Damaged Already. Will I Still Benefit from This Drug?

It is expected that CFTR modulator therapy is most effective if initiated early, before structural damage to the lungs has occurred. However, in the current practice, many patients starting ELE/TEZ/IVA therapy will already have longstanding structural lung damage. Although patients with severe lung damage resulting in a predicted FEV_1_ below 40% were not included in the phase III trials for the drug, there are clinical data available about the effects for this group of patients.

O’Shea et al. describe a group of 14 patients enrolled in the managed access program for ELE/TEZ/IVA, all with a ppFEV_1_ below 40 percent. At one month follow up, the mean ppFEV_1_ improved by 9.0%, and the mean BMI change was +1.7 kg/m^2^. Both results were statistically significant despite the small group size [18]. In the phase III trial published by Middleton et al., a subgroup was identified of 18 patients whose lung function declined to values below a ppFEV_1_ of 40% between inclusion and the start of treatment. After four weeks, the mean difference FEV_1_ was 15.2% compared to the placebo [13] The effects on lung function in this group are therefore comparable to the results in the phase III trials, in which patients were required to have ppFEV_1_ of 40–90% before the start of treatment. Both these studies were focused on short-term outcomes. A larger group of pwCF with advanced lung disease starting ELE/TEZ/IVA was evaluated by Burgel et al. The follow-up duration was 9 months, longer compared to the other studies. Gain in lung function was comparable to that in the phase III trials, and additional effects on the need for supplemental oxygen and decrease in enlistment for transplantation were reported [19].

There are no studies concerning the effects of ELE/TEZ/IVA on severely affected lungs in the longer term yet. For pwCF with moderately affected lungs (ppFEV1 40–90%), effects on lung function have been reported for the first 24 weeks of treatment, showing a mean increase in ppFEV_1_ of 11.9% for pwCF with the F508del/F508del genotype who received TEZ/IVA earlier, to 14.9% for pwCF with the F508del genotype and a minimal function mutation who received no prior CFTR-modulating treatment [20]. To get a glance at the longer-term outcomes in pwCF with advanced lung disease, it may be useful to look at the long-term effects of ivacaftor for patients with a gating mutation and severely affected lungs, as both ivacaftor in patients with a gating mutation and ELE/TEZ/IVA for patients with at least one F508del allele have a nearly similar effect in restoring CFTR function. Taylor-Cousar et al. describe a group of 44 patients with the F508del/G551D genotype and ppFEV_1_ < 40% or listed for lung transplantation. After 24 weeks of treatment, an increase in ppFEV_1_ of 5.5% was seen and an increase in weight of 3.3 kg [21]. Barry et al. describe 21 patients with the F508del/G551D genotype with a mean follow up of 237 days and ppFEV_1_ < 40% or listed for transplant, and found a FEV_1_ change of 4.2% along with a marked reduction in exacerbations [22]. A smaller group of 14 patients with a gating mutation and ppFEV_1_ < 40% was studied by Hebestreit et al. In this group, an improvement of ppFEV_1_ of 5.2% was seen after a mean follow up of 34 weeks [23].

In conclusion, there is a mean improvement in FEV_1_ after the start of highly effective CFTR modulation, even in pwCF with severely affected lungs. However, this change will be smaller compared to patients with mild or moderate lung damage. On other parameters such as BMI, the effects are similar even with advanced disease.

### 2.2. Is There Any Way to Know in Advance If My Response Will Be Better or Less Than Average?

To get an estimation of medication effects before starting the drug in real life, the in vitro measurement of CFTR function by the use of organoids can be useful. Berkers et al. describe the in vitro effect of ivacaftor and LUM/IVA on patient-derived rectal organoids that express CFTR abundantly, which correlates with the clinical effects of the tested CFTR modulators. In specific cases, such as patients with a rare genotype, this organoid technique might help to assess if a CFTR modulator will be effective or not [24]. A recent addition to CFTR testing in patient-derived cells is the assay on cultured nasal epithelial cells. Using this method, it would be harder to test cells of many different patients at the same time, but on an individual level, it is an accurate and feasible way to assess the effect of CFTR modulators in patients cells. The method was also validated for the correlation between the in vitro response and clinical effects of CFTR modulators [3].

### 2.3. Will I Gain Weight?

The effects of CFTR modulators on nutritional status have been reported in a review by Bailey et al. From available studies on the respective drugs, they found a relevant BMI increase with ivacaftor for pwCF with G551D, but not with R117H. The BMI increase with TEZ/IVA was not found to be significant. For the effect of ELE/TEZ/IVA on BMI, the review based itself of the phase III trials, finding a BMI change of +1.04 in pwCF heterozygous for F508del after 24 weeks of treatment, and +0.6 in pwCF homozygous for F508del after 4 weeks of treatment [25].

Since then, Griese et al. have added data for the longer term and found an increase in BMI of 1.30 compared to baseline for pwCF homozygous for F508del after 36 weeks, and an increase of 1.28 compared to baseline for pwCF heterozygous for F508del after 48 weeks of treatment [20].

### 2.4. Will There Be (Severe) Side Effects?

Based on reporting of adverse events during the phase III trials, side effects for ELE/TEZ/IVA are infrequent and usually mild. The main reported side effects directly attributed to ELE/TEZ/IVA use are rash and elevated transaminases. During the first 24 weeks of treatment, rash occurred in 9.9–10.9% of patients, leading to discontinuation in 0.2–0.5%. Elevated transaminases occurred in 4.2–5.4% of patients, leading to discontinuation in 0–0.6% [20]. Adverse events, such as hemoptysis or pulmonary exacerbations, occurred more often in the placebo group compared to the treatment group. Cough, however, was more common in the treatment group, as well as upper respiratory tract infections [12,13].

More severe adverse events have been reported to occur incidentally, mainly through case reports, with serum sickness-like reaction, biliary disease and testicular pain as examples [26,27,28]. No data are available so far about the use of ELE/TEZ/IVA in patients with severe liver failure due to CFLD.

### 2.5. Will This Treatment Save Me Some Hospital Admissions?

There are some available data on exacerbation rates and hemoptysis after the start of ELE/TEZ/IVA, so that an estimation of reduction in hospital admission should be feasible.

In the phase III study for this drug in F508del homozygous patients, exacerbation counts were low due to the short follow up of 4 weeks. In the ELE/TEZ/IVA group, exacerbations were reported in 2% of patients compared to 12% in the TEZ/IVA group, and hemoptysis in 4% versus 10% [12]. The other phase III trial that included patients with F508del and a minimal function mutation had a longer follow up time of 24 weeks. During this time, pulmonary exacerbations occurred in 21.8% of the ELE/TEZ/IVA group versus 47.3% in the TEZ/IVA group. Hemoptysis also occurred less in the ELE/TEZ/IVA group versus the TEZ/IVA group: 5.4% versus 13.9%, respectively [13].

In the prolonged phase III trial, including both F508del homozygous and compound heterozygous patients, the exacerbation rate was reported at 0.30 per 48 weeks, but this study had no control group and the exacerbation rate was not compared to that before treatment [20].

As infectious pulmonary exacerbations are the main cause for hospitalization for CF patients, it is very likely, based on the current evidence, that the use of ELE/TEZ/IVA will lead to a drastic reduction in hospital admissions.

### 2.6. Will This Drug Improve My Sinus Problems?

Chronic rhinosinusitis and nasal polyps contribute to the morbidity in people with CF, with a large impact on the quality of life. Description of the effects of ELE/TEZ/IVA on these sinonasal problems has been performed in two studies so far, both evaluating symptoms using the SNOT-22 questionnaire. DiMango et al. included 43 participants, and in this group, a reduction of the SNOT-22 score of 10.5 points was seen, from a baseline score of 34.8 [29]. Douglas et al. evaluated a group of 25 patients and found very similar effects; a score reduction of 10.2 from a baseline score of 34.5 [30].

As both studies used a validated symptom scale and found a large difference in symptom score after treatment, the conclusion should be that ELE/TEZ/IVA can significantly reduce sinonasal symptoms.

### 2.7. What Is Known about This Drug in the Context of Starting a Family?

As it is known, CFTR is expressed in the cervical and endometrial epithelium, causing decreased fertility in many females with cystic fibrosis; thus, it is logical that the use of a CFTR modulator can result in the increased incidence of pregnancy, which has also been described for ELE/TEZ/IVA in a case series [31]. This is, however, important to discuss before the start of treatment, because in the past with ivacaftor, this has resulted in unexpected and sometimes unwanted pregnancies. If a woman with CF gets pregnant while using CFTR modulators, it is important to weigh the benefits of the treatment to the mother versus the potential harm to the child. However, very limited data are available about the unborn child’s safety during the use of CFTR modulators in the mother [32].

One recent study is available on the safety of ELE/TEZ/IVA, including 45 patients who used the medication for a median duration of 3 months during pregnancy. During the study, 29 pregnancies resulted in live birth and 7 reached the second or third trimester without complications.

Four first trimester miscarriages occurred, which is in line with usual incidence of miscarriages in the US, of which one was reported as having unknown relatedness to modulator use, and the other three were deemed unrelated. Two unintentional pregnancies were electively terminated, and one first trimester pregnancy was electively terminated because of severe malformations in a woman with poorly regulated diabetes. No exacerbation deemed related to ELE/TEZ/IVA use was reported. Five premature births occurred, none of which were deemed related to the medication. In the infants, screening for cataracts was not standard. In the four infants who underwent screening, no cataracts were found. It is known that all components of ELE/TEZ/IVA, and also of LUM/IVA, are present in breast milk if the mother takes this medication. Screening of the child for cataracts is not always done, and in the two children who were screened for this indication, no cataracts were found [33]. One child who was breast fed while the mother was on LUM/IVA therapy had elevated transaminases by day 29 after birth, which normalized after reduction of the breast-feeding fraction and recurred with full breast feeding. Although this was just one reported case, it could be advisable to monitor transaminases in breastfed infants whose mother takes CFTR modulators [34].

### 2.8. Will I Be Able to Do Sports/Work Again?

Although the effects of ELE/TEZ/IVA on lung function are convincing for its effectiveness, the exact effects on daily functioning will be hard to predict, mainly because mentioned work or sports activities will vary a great deal between individual, as will their baseline condition before the start of treatment.

To have some estimation of functional gain, CFQ-R scores before and after the start of therapy can be helpful. However, most studies only report the respiratory domain, although the domains ‘Role’, ‘Social’, ‘Vitality’ and ‘Physical functioning’ will better help estimate functional gain. One study reported all CFQ-R domains (scale 0–100) for a group of 43 patients before treatment and after three months. In the ‘Social’ domain, the mean change in scores was +6.7. Scores in ‘Role’, ‘Vitality’ and ‘Physical’ functioning improved by 10.0, 12.5 and 13.3 points, respectively. A minimal clinically important difference was reported in the literature for the respiratory domain only [35].

## 3. Discussion

When starting a new CFTR modulating drug, it is of great importance to discuss expectations about the new treatments in an open manner. This helps the patient to have a realistic view of the potential effects for their situation, but also knowing about possible adverse effects will improve safety of the treatment. This review could support that by providing an overview of the now available literature. However, it will not be able to answer all individual patient questions for multiple reasons.

At this point in time, data about the effects of ELE/TEZ/IVA are very limited on all aspects of treatment, although new information is published at a high speed. In the upcoming years, there will at least be a lack of long-term data about this treatment.

In some situations where the literature about ELE/TEZ/IVA effects is lacking, it can be a good option to turn to studies performed with IVA, which also has a potent CFTR-modulating effect. It is important when reviewing this evidence that these studies specifically included patients with a gating mutation, who on average will have a better condition compared to patients with class I or II mutations at the same age. Moreover, the pharmacokinetic and pharmacodynamic properties of the combination drug are different from ivacaftor alone. Therefore, extrapolation of study results about other CFTR modulators should be done with caution only, even when it is the best information available at the time.

To be able to inform an individual patient about the expected effects, it is not only important to have recent literature available, but also to evaluate if your patient’s situation is comparable to that of the study population.

## 4. Conclusions

The current literature points towards favorable effects of ELE/TEZ/IVA on lung function as well as sinus problems and quality of life. The drug is overall well tolerated and severe adverse reactions are scarce. ELE/TEZ/IVA seems to improve fertility in female pwCF, while the treatment’s continuation during pregnancy is currently advised against, due to a lack of robust data about risks to the unborn child. Cell models such as organoids or cultured nasal cells can be used to estimate in advance if effectivity of the drug can be expected. An increasing amount of data are published regarding the effects of ELE/TEZ/IVA, helping to better inform patients in the discussion of their expectations when starting new CFTR modulators. However, the literature addressing differences in the response between patient groups with selected characteristics is still very limited.

## Data Availability

Not applicable.

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
