# Peer review of "Management of Individual Patient Expectations When Starting with Highly Effective CFTR Modulators"

_jpm, 2021, doi:10.3390/jpm11080811_

Round 1

Reviewer 1 Report

Aalbers, et al. reviewed recent literature on various aspects of CFTR modulator therapy based on patient questions/expectations. The topic is relevant to CF care in the post-modulator era. The way the authors arranged the clinical information accordingly to various commonly asked questions is innovative. The major weakness of the manuscript is the authors' failure to summarize their specific findings on the subject in the abstract. Instead, they just described what they did.

Author Response

We have adjusted the abstract to make it more of a summary rather than an introduction. Changes can be found on page 1 lines 7-23.

If any other concerns arise, we’d be glad to learn about these so we can adjust the manuscript.

Reviewer 2 Report

All of my concerns from the previous revision, have been addressed. I recommend to accept it in present form.

Author Response

Based on the other reviewers’ comments, we have adjusted the abstract. In the rest of the manuscript, no changes were made. If any new concerns arise, we’d be glad to learn about this.
